# Hydrothermal Synthesis of Layered Double Hydroxides Doped with Holmium, Thulium and Lutetium

Sergei N. Golovin *, Maksim N. Yapryntsev and Olga E. Lebedeva

Institute of Pharmacy, Chemistry and Biology, Belgorod State National Research University,
308015 Belgorod, Russia
* Correspondence: 801492@bsu.edu.ru

**Abstract:** For the first time, nickel-aluminum layered double hydroxides containing holmium, thulium or lutetium cations were successfully synthesized via the coprecipitation method followed by hydrothermal treatment. X-ray diffraction data show the formation of hydrotalcite-like structures with the absence of impurity phases. The presence of lanthanides is confirmed by energy-dispersive X-ray spectrometry. Assumed empirical formulae for these compounds are $[Ni_{0.796}Al_{0.193}Ho_{0.011}][(NO_3)_{0.204}\cdot yH_2O]$, $[Ni_{0.808}Al_{0.178}Tm_{0.015}][(NO_3)_{0.193}\cdot yH_2O]$ and $[Ni_{0.765}Al_{0.219}Lu_{0.016}][(NO_3)_{0.235}\cdot yH_2O]$. These novel layered compounds demonstrate the fundamental ability of doping hydrotalcite-like compounds with any lanthanide cation, which gives rise to the synthesis of new materials with specific properties.

**Keywords:** layered double hydroxide; hydrothermal synthesis; holmium; thulium; lutetium





## 1. Introduction

Layered double hydroxides (LDHs) are either natural or synthetic inorganic substances, which consist of positively-charged brucite-like octahedral layers, alternating with interlayers of compensating anions and water molecules. Since the first discovered layered double hydroxide was the mineral hydrotalcite $[Mg_6Al_2(OH)_{16}](CO_3)\cdot4H_2O$, these compounds are often called hydrotalcite-like. The general formula for LDHs is $[M(II)_{1-x}M(III)_x(OH)_2]^{x+}[A^{n-}_{x/n}\cdot yH_2O]^{x-}$, where M(II) and M(III) are divalent and trivalent metal cations, respectively, and $A^{n-}$ is n-charged anion. The number of different variations of such layered structures is great and depends on the nature of cations, the molar ratio of double-charged and triple-charged cations, and the type of anions [1].

A wide set of cations, the possibility of intercalation of various anions or anionic complexes, as well as the regulation of the M(II)/M(III) ratio reveal great opportunities for obtaining multifunctional materials with predetermined properties. It leads to a broad range of applications for layered double hydroxides and their derivatives. Promising routes for their potential using are heterogeneous catalysis [2], wastewater clean-up [3], targeted drug delivery [4], electrodes for supercapacitors [5].

Doping of LDHs with rare earth elements, particularly lanthanides, seems to be a promising task, since they are known for their pronounced optical [6], catalytic [7] and electric [8] properties. To date, lanthanide-containing layered double hydroxides have been studied incompletely and unevenly. The majority of works are devoted to studying cerium-, europium- and terbium-containing ones, while others are represented very sparsely [9] (p. 2). Moreover, we cannot find any information about synthesis and investigation of LDHs doped with holmium and thulium cation. Only one article is devoted to the attempt of lutetium doping of Ni/Al-LDH, which was unsuccessful [10]. The authors assume that lutetium cations could not be incorporated into hydroxide layers due to their larger ionic radius. However, other lanthanides have radii close to $Lu^{3+}$ and are quite capable of substituting aluminum ions in nickel-aluminum-layered double hydroxides [11,12]. So, the objective of the actual research was an attempt to synthesize Ni/Al-LDHs containing

holmium, thulium and lutetium cations to expand the range of materials with a hydrotalcite-like structure.

## 2. Results and Discussion

Figure 1 shows the X-ray powder diffraction (XRPD) patterns of the obtained materials. According to X-ray analysis results, only one phase with a hexagonal structure (space group R$\overline{3}$m, № 166) is detected in studied samples. All reflexes are in good agreement with the standard data for jamborite (ICDD (PDF2.DAT) № 01-089-7111). Thus, there is a reason to consider that lanthanides successfully incorporated into octahedral layers. Although the presented diffractograms look quite typical for hydrotalcite-like compounds, reflections are strongly broadened and the ratio of reflection intensities differs from the most common pattern [1] (Figure 10a). Such broadening may be due to insufficient aging time or it may be the effect of doping since lanthanide cations are larger than aluminum ones. Moreover, layered double hydroxides often exhibit polytypism as well as different kinds of structural disorder, for example, interstratifications, turbostraticity and stacking faults, which have a great impact on diffraction pattern. In our case, diffractograms have features similar to those received by simulating these effects [13] (Figure 6), [14] (Figure 6), [15] (p. 12884).

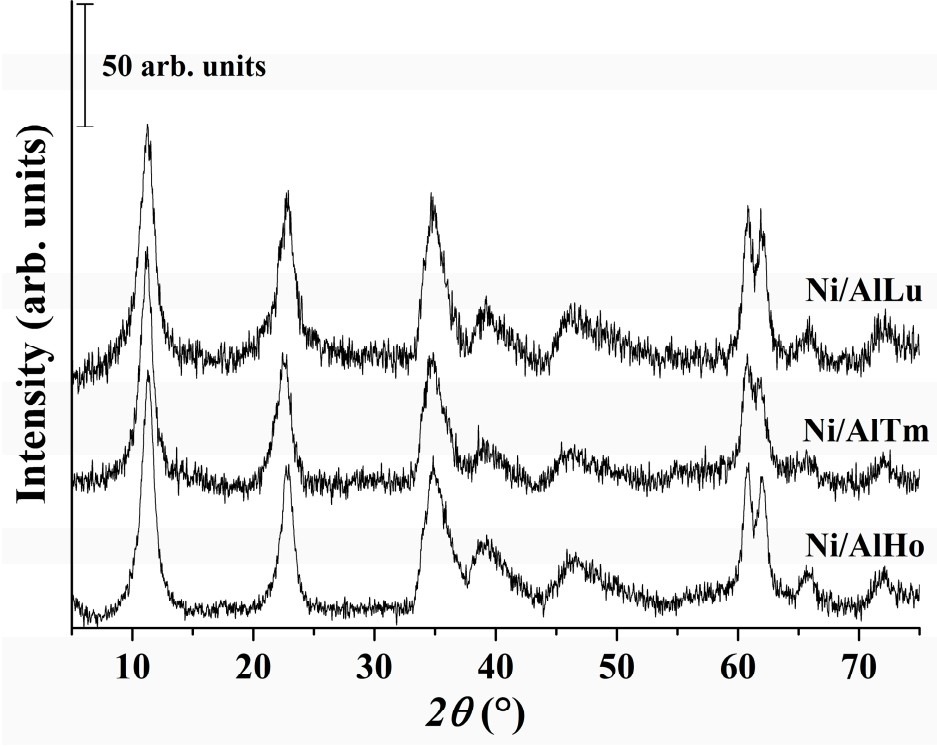

**Figure 1.** XRPD pattern of as-synthesized lanthanide-containing layered double hydroxides.

The presence of lanthanide cations is confirmed by energy-dispersive X-ray spectrometry (EDX) (Figure 2). Nevertheless, resulting cationic molar ratios do not match to the predetermined values (Table 1). Apparently, this is due to drawbacks of synthesis technique. The precipitation occurred under conditions of high supersaturation, which often results in a product with incorrect $M^{2+}/M^{3+}$ ratio [16] (pp. 95–97). Assumed empirical formulae for these compounds are $[Ni_{0.796}Al_{0.193}Ho_{0.011}]$ $[(NO_3)_{0.204} \cdot yH_2O]$, $[Ni_{0.808}Al_{0.178}Tm_{0.015}]$ $[(NO_3)_{0.193} \cdot yH_2O]$ and $[Ni_{0.765}Al_{0.219}Lu_{0.016}]$ $[(NO_3)_{0.235} \cdot yH_2O]$. Carbonates, which are likely to settle in the interlayer space during the synthesis, were not analyzed.

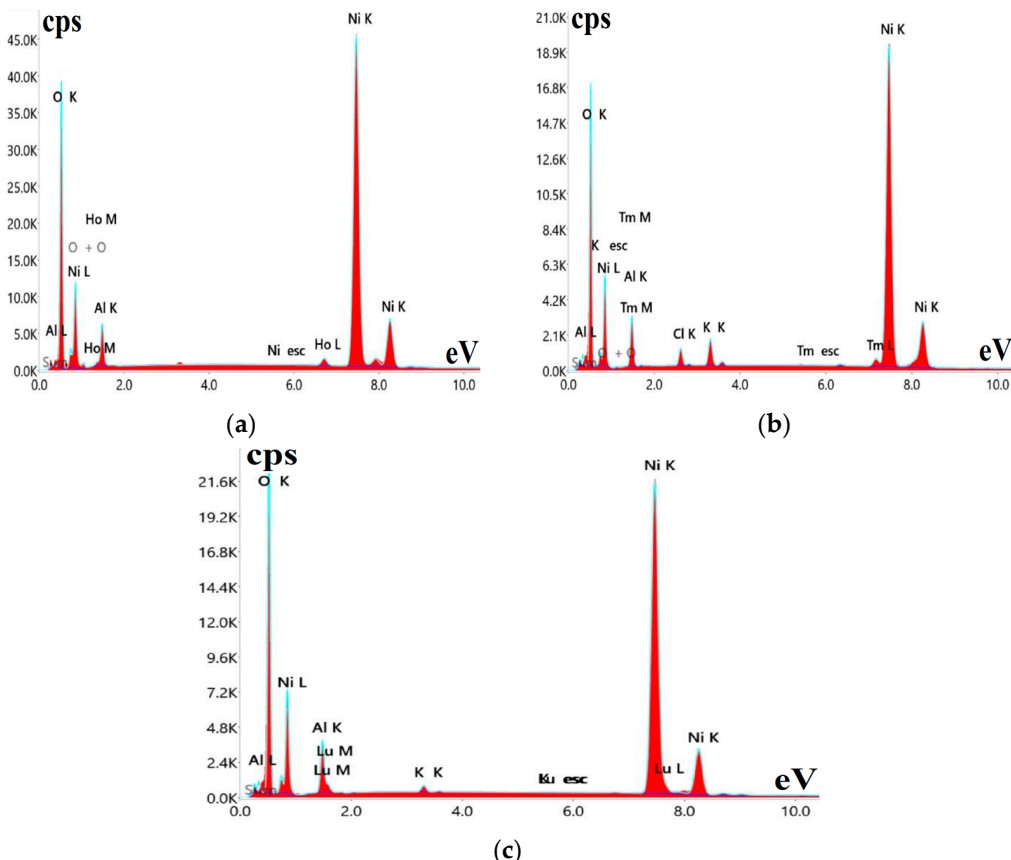

**Figure 2.** EDX spectra of as-synthesized lanthanide-containing layered double hydroxides: (**a**) Ni/AlHo; (**b**) Ni/AlTm; (**c**) Ni/AlLu.

**Table 1.** Cationic molar ratios and lattice parameters of as-synthesized lanthanide-containing layered double hydroxides.

| Sample | Molar Ratios of Cations | | | $M^{2+}/M^{3+}$ | $Ln^{3+}/(Ln^{3+} + Al^{3+})$ | Lattice Parameters | | Average Crystallite Size *, nm |
|---|---|---|---|---|---|---|---|---|
| | $Ni^{2+}$ | $Al^{3+}$ | $Ln^{3+}$ | | | $a$, Å | $c$, Å | |
| Ni/AlHo | 0.796 | 0.193 | 0.011 | 3.9 | 0.054 | 3.103 | 23.39 | 6.67 |
| Ni/AlTm | 0.808 | 0.178 | 0.015 | 4.2 | 0.078 | 3.056 | 23.72 | 6.89 |
| Ni/AlLu | 0.765 | 0.219 | 0.016 | 3.3 | 0.068 | 3.049 | 23.59 | 5.56 |

\* calculated using the Scherrer equation.

Determined lattice parameters are presented in Table 1. The value of parameter *a* can be quite informative, since it corresponds to cation-cation distance in octahedral layers and, therefore, depends on cationic radii. Cationic radii of $Ho^{3+}$, $Tm^{3+}$ and $Lu^{3+}$ are close to each other and almost double the radius of $Al^{3+}$ [17] (pp. 305, 307). So, the incorporation of lanthanide ions should lead to an increase in interion distance within LDH layers and consequently parameter *a*. Parameter *c* is less important in our case because it mainly depends on electrostatic interaction between hydroxide layers and interlamellar anions and corresponds to the thickness of the sheets. In the study [18] (p. 29) for of Ni/Al-$CO_3$ LDH with $Ni^{2+}/Al^{3+}$ = 3.01 and 4.09 Kovanda et al. received values of *a* equal to 3.042 Å and 3.050 Å, respectively. Although a greater difference was expected, these values are still slightly less than ours. Perhaps, the amount of doping element is not enough to significantly increase cation-cation distance. At the same time, for the parameter *a* of Ni/Al-$NO_3$ LDH with $Ni^{2+}/Al^{3+}$ = 3 Wang et al. [19] (p. 7213) give value 3.01 Å, which is much less. Thus, based on the combination of XRPD and EDX data, we assert about the

synthesis of single-phase layered double hydroxides containing holmium, thulium and lutetium cations.

The morphology of synthesized materials was investigated with transmission electron microscopy (Figure 3), which showed that they consist of plate-like particles with a shape similar to hexagonal. The magnification factor was 150,000 for Ho- and Tm- and 200,000 for Lu-containing samples. The average particle size for all samples is very close and approximately equals to 25 nm for Ni/AlHo- and Ni/AlLu-LDHs and 24 nm for Ni/AlTm-LDH (determined using Digimizer software). Such morphology is common among layered double hydroxides and was observed earlier [18] (Figure 4d), [20] (Figure 2b,c), including our works [21] (Figure 6).

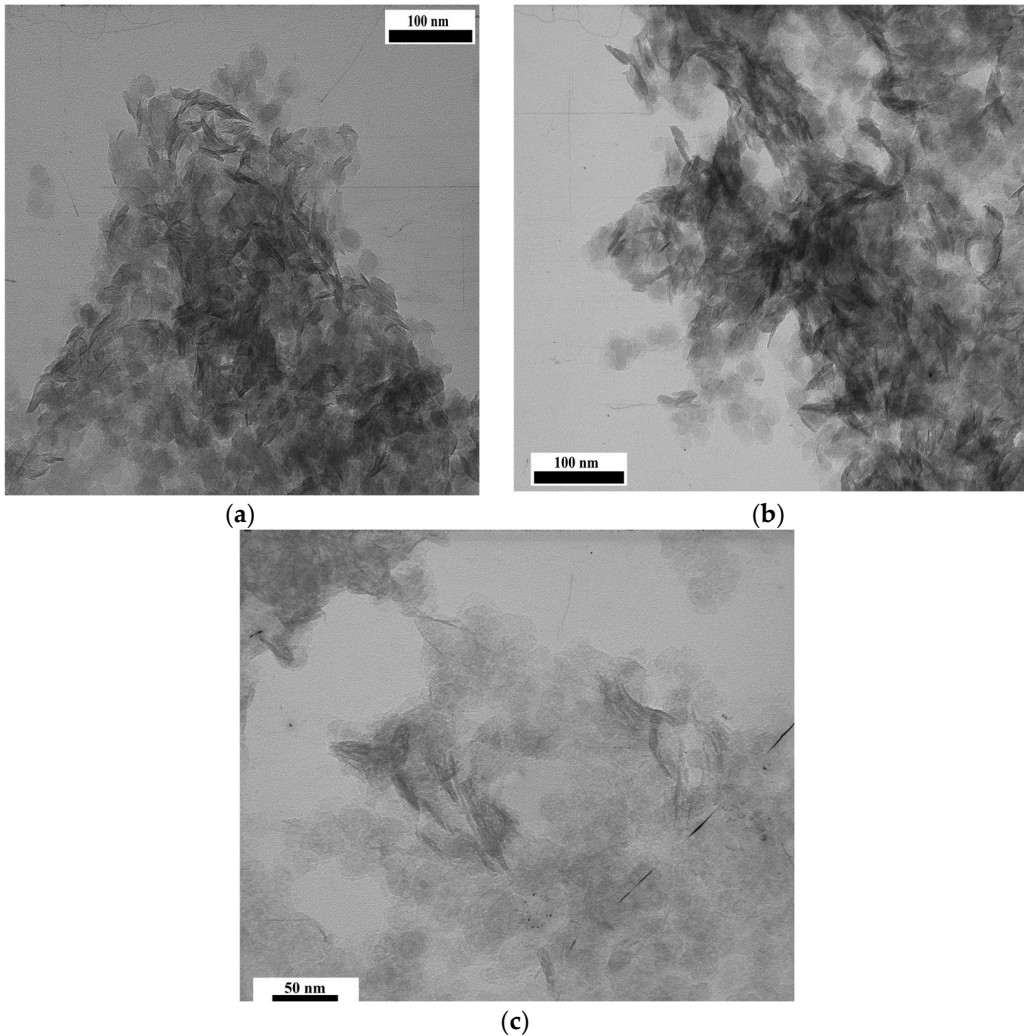

**Figure 3.** TEM images of as-synthesized lanthanide-containing layered double hydroxides: (**a**) Ni/AlHo; (**b**) Ni/AlTm; (**c**) Ni/AlLu.

## 3. Materials and Methods

Nickel-aluminum layered double hydroxides doped with holmium, thulium and lutetium were synthesized by the coprecipitation method followed by hydrothermal treatment. Corresponding nitrates were used as cation sources and aqueous solution of potassium hydroxide KOH as precipitating agent. Predetermined values of cationic molar ratios were as follows: $M^{2+}/M^{3+} = 3$ and $Ln^{3+}/(Ln^{3+} + Al^{3+}) = 0.05$, where $Ln^{3+}$ is lanthanide cation. A precipitant solution ($C_M = 0.625$ mol·$L^{-1}$) was added dropwise under stirring to the aqueous nitrate solution with total cationic concentration of 0.25 mol·$L^{-1}$. Then the mixture was transferred into 50 mL autoclave reactor (Parker autoclave Engineers) and hy-

drothermally aged for 6 h at 120 °C. The precipitate was centrifuged from the mother liquor, washed with distilled water until pH $\approx$ 7 and air dried at the temperature of 20–25 °C.

The phase composition of obtained bulk samples was examined by X-ray powder diffraction (XRPD). XRPD patterns were received on Rigaku Ultima IV diffractometer using CuK$_\alpha$ radiation ($\lambda$ = 1.54056 Å). The record was carried out at a rate of 2°/min and steps of 0.02 in the range 2θ = 5°–75°. PDF database was used for the identification of reflexes. Lattice parameters were determined by PDXL software. The elemental composition of obtained materials was studied by the energy-dispersive X-ray spectrometry (EDX) with QUANTA 200 3D scanning electron microscope equipped with an energy-dispersive analyzer at the operating voltage of 30 kV. The morphology of the samples was investigated by transmission electron microscopy (TEM) with JEM-2100 at 200 kV with 0.2 nm resolution. Sample preparation for TEM research was as follows. Thin carbon film sprayed on a copper grid and covered with a drop of the suspension of the sample in anhydrous isopropyl alcohol, which was ultrasonicated for 30 min. The prepared single-layer non-self-supporting sample was air-dried to remove isopropyl alcohol for 30 min.

## 4. Conclusions

We report the synthesis of nickel-aluminum-layered double hydroxides containing holmium, thulium or lutetium cations by coprecipitation method followed by hydrothermal treatment. As far as we know, doping with Ho$^{3+}$ and Tm$^{3+}$ was not performed earlier and doping with Lu$^{3+}$ was unsuccessful. XRPD analysis indicates no evident impurities in samples and EDX data confirms the presence of lanthanides. The morphology of these materials is similar and typical for LDHs.

**Author Contributions:** Conceptualization, O.E.L.; methodology, O.E.L.; software, S.N.G. and M.N.Y.; validation, S.N.G., M.N.Y. and O.E.L.; formal analysis, O.E.L.; investigation, S.N.G. and M.N.Y.; resources, M.N.Y.; data curation, O.E.L.; writing—original draft preparation, S.N.G.; writing—review and editing, O.E.L.; visualization, S.N.G.; supervision, O.E.L.; project administration, O.E.L.; funding acquisition, O.E.L. All authors have read and agreed to the published version of the manuscript.

**Funding:** This research was funded by Russian Foundation for Basic Research, grant number 20-33-90178.

**Institutional Review Board Statement:** Not applicable.

**Informed Consent Statement:** Not applicable.

**Data Availability Statement:** Not applicable.

**Acknowledgments:** The work was carried out using the equipment of the Joint Research Center of Belgorod State National Research University «Technology and Materials».

**Conflicts of Interest:** The authors declare no conflict of interest. The funders had no role in the design of the study; in the collection, analyses, or interpretation of data; in the writing of the manuscript; or in the decision to publish the results.

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
