# Peer review of "Hydrothermal Synthesis of Layered Double Hydroxides Doped with Holmium, Thulium and Lutetium"

_inorganics, doi:10.3390/inorganics10120217_

Round 1
Reviewer 1 Report
The following manuscript entitled ‘Hydrothermal synthesis of layered double hydroxides doped
with holmium, thulium and lutetium’ authored by Golovin et. al. is a nice and novel work. The authors were able to hydrothermally synthesize three lanthanide (Ho, Tm and Lu) doped layered double hydroxides followed by characterization with PXRD, EDX and TEM experiments. Overall this is a good work but I think there is scope to modify that would make the manuscript appear more impressive.
Comments/questions:
-
The composition of the synthesized compounds can be written in the abstract matching with the general empirical formula of this kind of double layered hydroxides as mentioned in the introduction section.
-
Figure 3 shows the TEM image of the synthesized Ni/AlLu compound, did the other two compounds also show similar results or what is the reason for not checking them?
-
In the synthetic procedure the temperature of drying can be mentioned instead of saying ambient temperature, so that the results can be reproducible by any other chemists/scientists who will be interested. What was the yield for these synthesis?
-
The double layered hydroxides doped with other lanthanides have already been reported in the literature. Did the authors notice any unusual behavior of their compounds from other reported compounds of this type? It would be good if the authors mention somewhere how these compounds can be utilized in real application or what is the takeaway from this research.
Reviewer 2 Report
Article by S.N. Golovin, Maxim N. Yapryntsev and O.E. Lebedev "Hydrothermal synthesis of layered double hydroxides doped with holmium, thulium and lutetium" devoted to a new synthesis of nickel-aluminum layered double hydroxides containing holmium, thulium or lutetium cations by co-precipitation method followed by hydrothermal treatment. The manuscript is presented as Communication.
The main remarks are given below.
1. All diffractograms in Fig. 1 look the same. How do they give information about which particular lanthanide has been introduced? The method is not informative or requires more detailed discussion. The diffractograms themselves are not complete enough. The card numbers from the Powder X-Ray Phase Analysis Database are not marked, or should the compounds be entered into this Database? If we are talking about nanoparticles, then it is possible to calculate their size using the Debye-Scherrer formula.
2. The authors directly refers to the discrepancy between the molar ratios of cations that are not given values (table 1), which is most likely due to a poorly designed experiment. Such things should be checked for reproducibility.
3. In Fig. 2 (EDX analysis data) the scales are not labeled and the eV values for the elements are not described in the text.
4. The only TEM image for sample with lutetium is shown, although the article is devoted to the synthesis of layers of double hydroxides doped with holmium, thulium and lutetium. TEM images of the two remaining hydroxides should also be provided to determine (or validate) their size and shape. The article looks incomplete without this data.
5. The article lacks Conclusions sections, where all data on new synthesized compounds were briefly summarized. The article looks unfinished and cannot be accepted for publication in this form.
Reviewer 3 Report
Manuscript: Hydrothermal synthesis of layered double hydroxides doped with holmium, thulium and lutetium
In this manuscript, authors presented successful attempt to synthesize Ni/Al-LDHs containing holmium, thulium and lutetium cations to expand the range of materials with hydrotalcite-like structure. The idea is very interesting and presentation of work is generally good. However, there are some changes authors should make before manuscript can be accepted for publication in Inorganics journal.
1. Authors should put one sentence which implies the significance of obtained results in the end of the Abstract.
2. Please, put more relevant references into Introduction section. For example, (Line 25, Lines 26-29, Lines 34-35, etc.)
3. I think that abbreviation XRPD is more convenient than PXRD you used.
4. Figure 1. Instead of 2θ (degree), put 2θ (°).
5. Why did not you provide TEM micrographs for all synthesized samples? Also, the details about used magnification are missing. A presented micrograph is of low-resolution making impossible to corroborate some statements in the Discussion.
Round 2
Reviewer 2 Report
Author's responce was quet satisfactory. TEM imagines was added to the Manuscripte.
The only note that most of references is older than five years, but it is very often for scientific developments that early investigations is develope due to equipment and technology enhancements.